# Loss-of-function mutation of NSD2 is associated with abnormal placentation accompanied by fetal growth retardation in mice

Eriko Ohnishi[1¤], Shiori Kinoshita[1], Kazuhiko Nakabayashi[1], Kenichiro Hata[1,2*], Tomoko Kawai[3*]

**1** Department of Maternal-Fetal Biology, National Research Institute for Child Health and Development, Tokyo, Japan, **2** Department of Human Molecular Genetics, Gunma University Graduate School of Medicine, Gunma, Japan, **3** Division of Fetal Development, National Research Institute for Child Health and Development, Tokyo, Japan

¤ Current Address: Departments of Neurosurgery, Graduate School of Medicine, Yamaguchi University, Ube, Yamaguchi, Japan

* khata@gunma-u.ac.jp (KH); kawai-tm@ncchd.go.jp (TK)

## Abstract

Nuclear receptor-binding SET domain-containing 2 (NSD2) is the primary enzyme responsible for the dimethylation of histone H3 lysine 36 (H3K36me2), a marker associated with active gene transcription and intergenic DNA methylation. NSD2 is required for the normal development of humans and mice; however, its function during placentation remains unknown. Using genome editing techniques, we previously established two lines of *Nsd2*-mutant mice that showed growth retardation and neonatal lethality. Here, we further demonstrated that the loss-of-function mutation of NSD2 caused enlargement of the mouse placenta with morphological changes during late-gestation. *Nsd2*-mutant placentas were significantly heavier and showed thicker fetal layers with an expanded junctional zone and dilatated maternal blood sinuses in the labyrinth compared to their wild-type littermates. Abnormal placentation was accompanied by fetal growth defects, some with edema and one with a congenital cardiovascular anomaly, which may have partially affected neonatal survival. To our knowledge, this is the first study demonstrating the physiological and pathological functions of NSD2 during placentation.

## Introduction

The placenta is a vital organ for fetal development. Problems with its formation and function underlie many aspects of early pregnancy loss and complications in humans [1,2]. Large-scale screening of embryonic lethal and sub-viable mouse knockout lines revealed the crucial role of placental defects in abnormal embryonic development [3]. The dynamic placentation processes involve significant alterations in the gene expression profiles of endometrial and trophoblast lineages. Epigenetic control

**Data availability statement:** All relevant data are within the manuscript and its Supporting Information files.

**Funding:** This study was supported by: - Japan Agency for Medical Research and Development, (ek0109489), Dr. Kenichiro Hata; - Japan Agency for Medical Research and Development, (ek0109205), Dr. Tomoko Kawai; - Japan Society for the Promotion of Science, (21K19584), Dr. Kenichiro Hata; - National Center for Child Health and Development (JP), (2022A-3), Dr. Kenichiro Hata; - National Center for Child Health and Development (JP), (2024B-5), Dr. Tomoko Kawai; - Japan Society for the Promotion of Science, (22K11793), Dr. Tomoko Kawai.

**Competing interests:** The authors have declared that no competing interests exist.

mechanisms, including DNA methylation and histone post-translational modifications, act as regulatory switches that modulate gene activity. Emerging evidence has underscored the critical roles of histone modifications in such dynamic processes [4].

The nuclear receptor–binding SET domain (NSD) family proteins, including NSD1 (KMT3B), NSD2 (WHSC1/MMSET), and NSD3 (WHSC1L1), are methyltransferases that generate nucleosomes modified with H3K36me2 *in vitro*. Although the dysregulation of NSD proteins is associated with developmental defects and cancers [5], little is known about the functions of NSD proteins in the placenta. A recent study characterized the global change in histone H3 lysine modifications during the conversion of porcine-induced pluripotent stem cells (iPSCs) into trophoblast-like stem cells and found that the level of H3K36me2, but not other histone modifications, was higher in trophoblast-like stem cells than in iPSCs [6]. These findings suggest a potential role for NSD proteins in placental development.

NSD1 and NSD2 are required for normal development in humans and mice [5]. NSD1 is essential for early post-implantation development. *Nsd1*-deficient mice generated by gene disruption fail to complete gastrulation [7]. Although *Nsd1* is ubiquitously expressed in the ectoplacental cone and extraembryonic and embryonic germ layers of embryos on embryonic day 7.5 (E7.5) [7], the early embryonic lethality phenotype of *Nsd1*-deficient mice impairs the functional characterization of NSD1 during placental development. Using genome editing techniques, we previously generated two mouse lines: an *Nsd2*-knockout line created by completely deleting *Nsd2* genes (*Nsd2*$^{-/-}$) [8,9], and a knock-in line carrying a patient-derived single nucleotide point mutation that results in a proline-to-leucine substitution at position 906 of NSD2 (NP_001074571.2) (*Nsd2*$^{P906L/P906L}$) [8]. We demonstrated that this variant was pathogenic and showed that it destabilizes the NSD2 protein [8]. This finding aligns with recent research showing that 60% of pathogenic missense variants, among more than 500,000 variants across over 500 human protein domains, reduce protein stability [10]. Both *Nsd2*$^{-/-}$ and *Nsd2*$^{P906L/P906L}$ reduced H3K36me2 level, which accompanies DNA hypomethylation [8]. The genetic differences between *Nsd2*$^{-/-}$ and *Nsd2*$^{P906L/P906L}$ include the production of diverse mRNA isoforms of *Nsd2* and the expression of non-coding RNAs present in the deleted region in *Nsd2*$^{-/-}$. All *Nsd2*$^{-/-}$ and the majority (66.7%) of *Nsd2*$^{P906L/P906L}$ neonates died immediately after birth. We have also demonstrated that *Nsd2*$^{-/-}$ and *Nsd2*$^{P906L/P906L}$ embryos at E18.5 were growth-retarded compared to their wild-type littermates. The body weight at E18.5 was significantly reduced in *Nsd2*$^{-/-}$ and *Nsd2*$^{P906L/P906L}$ embryos [8,9].

Dysfunction of NSD1 or NSD2 results in global DNA hypomethylation in patients' blood cells [8,11]. DNA methylrtransferase (DNMT) contains histone-interacting domains. The proline-tryptophan-tryptophan-proline (PWWP) domain of DNMT3A recognizes both H3K36me2 and H3K36me3 in vitro with a higher affinity towards H3K36me2. NSD1/NSD2-deposited H3K36me2 has been shown to recruit DNMT3A and maintain DNA methylation in intergenic regions [12,13]. Therefore, NSDs likely play crucial roles in early development through multiple pathways of epigenetic regulation. All *NSD* family genes are expressed in placenta [14]. In contrast to the *Nsd1*-deficient mouse line [7], the perinatal lethality phenotypes of the *Nsd2*-mutant

mouse lines allowed us to characterize the consequences of NSD2 deficiency on placental development. Here, we assessed the fetal and placental phenotypes of $Nsd2^{-/-}$ and $Nsd2^{P906L/P906L}$ lines to elucidate the effects of NSD2 deficiency or impairment on placental development.

## Materials and methods

### Mice lines

As described previously, $Nsd2$-knockout ($Nsd2^{-/-}$) and $Nsd2$-knockin ($Nsd2^{P906L/P906L}$) mouse lines were generated using Clustered Regularly Interspaced Short Palindromic Repeat (CRISPR)-associated protein 9 genome editing [8]. All the mice used in this study were maintained on a C57BL/6 background. The C57BL/6J mice were obtained from CLEA Japan (Tokyo, Japan). This study was carried out in strict accordance with the recommendations of the Guidelines for the Care and Use of Laboratory Animals of the National Research Institute for Child Health and Development. All animal procedures were approved by the Institutional Animal Care and Use Committee (Permit Numbers: A2016-001). All dissections were performed under isoflurane anesthesia, making all efforts to minimize animal suffering.

### Mouse phenotyping and statistical analysis

The mice were housed under standard laboratory conditions with controlled temperature ($23 \pm 1°C$), relative humidity ($50 \pm 20\%$), and photoperiod (12-hour light/dark cycle: light on at 8:00 and off at 20:00). For each $Nsd2$-KO and $Nsd2$-KI line, heterozygous male and female individuals were mated to obtain embryos of the three genotypes. Individual females were examined for vaginal plugs to determine whether mating had occurred. Pregnant female individuals were euthanized by cervical dislocation under isoflurane anesthesia to obtain embryos at E15.5 and E18.5. The fetus and placenta were separated from each embryo and weighed. The genotype of each embryo was determined as previously described [8]. A chi-square goodness-of-fit test using Mendelian ratios was performed to evaluate genotypic ratios. Relative body weight, placenta weight, and placenta/body weight ratio were determined as ratios to the mean of the control group. Statistical tests for the three genotypes were performed by pairwise comparison using non-paired Wilcoxon rank sum test with Holm's correction for multiple testing. A value of $p < 0.05$ was considered statistically significant.

### Histological analysis

Fetal hearts and placentas were dissected at E15.5 and fixed by immersion in 10% formalin neutral buffer solution (FUJIFILM Wako Pure Chemical Corporation, Osaka, Japan). Tissue samples were generated using an automated tissue processor (Leica ASP 200; Leica Biosystems, Wetzlar, Germany), embedded in paraffin, and sectioned using a rotary microtome. Serial heart sections (4 μm) were cut and stained with hematoxylin and eosin (HE). The slides were scanned using the BZ-X800 imaging software (Keyence Corporation, Osaka, Japan). Serial placental sections (5 μm) were stained with HE for general morphology and periodic acid-Schiff (PAS) which labels glycogen, glycoproteins, and glycolipids. Mayer's hematoxylin solution was used for counterstaining. The slides were scanned using an Aperio AT2 whole digital scanner (Leica Biosystems, Vista, CA, USA); images were obtained and analyzed using an Aperio ImageScope (Leica Biosystems, Vista, CA, USA).

### Histochemical analysis of alkaline phosphatase activity

E15.5 placentas were fixed in 4% paraformaldehyde solution (MUTO PURE CHEMICALS CO., Tokyo, Japan) overnight at 4°C and processed for paraffin embedding. Serial 6-μm sections were dewaxed and incubated in BCIP-NBT Solution (nacalai tesque, Kyoto, Japan) for 4 hours at room temperature in the dark for endogenous alkaline phosphatase activity detection. The slides were scanned using the BZ-X700 imaging software (Keyence Corporation, Osaka, Japan).

## Results

### Fetal development in *Nsd2* mutant mice

To confirm the effects of loss-of-function mutation in NSD2 on development, we examined the genotypic ratios of progenies from intercrosses between heterozygous mutant mice in each *Nsd2*-knockout and -knockin line (Table 1). In *Nsd2*$^{+/-}$ intercrosses, both *Nsd2*$^{-/-}$ and *Nsd2*$^{+/-}$ embryos were present at the expected frequencies at both E15.5 and E18.5 stages. Embryonic lethality was observed at E15.5 in one out of 43 *Nsd2*$^{+/-}$ (2.3%) and one out of 16 *Nsd2*$^{-/-}$ (6.3%) embryos but not in the *Nsd2*$^{+/+}$. No embryonic lethality was observed at E18.5 in *Nsd2*$^{+/-}$ intercrosses. In *Nsd2*$^{WT/P906L}$ intercrosses, embryonic lethality was observed in one out of 34 *Nsd2*$^{WT/WT}$ embryos (2.9%) at E15.5 and in three out of 61 *Nsd2*$^{WT/P906L}$ (4.9%) and one out of 19 *Nsd2*$^{P906L/P906L}$ (5.3%) embryos at E18.5. The surviving number of *Nsd2*$^{WT/P906L}$ and *Nsd2*$^{P906L/P906L}$ embryos tended to be less than the expected Mendelian ratios, especially at E18.5, but were not statistically significant (Table 1). Hence, a loss-of-function mutation in NSD2 is not necessarily lethal during embryogenesis. Resorption was detected more frequently in heterozygous progeny of both *Nsd2*-mutant lines, but the possibility of contamination by maternal tissue during dissection cannot be ruled out. A more detailed analysis is needed for an accurate assessment.

**Table 1. Genotypic analysis of the progeny from *Nsd2* heterozygous intercrosses.**

| *Nsd2*$^{WT/-}$ intercross at E15.5 | *Nsd2*$^{+/+}$ | *Nsd2*$^{+/-}$ | *Nsd2*$^{-/-}$ | Total | p-value |
|---|---|---|---|---|---|
| Total embryos | 16 | 43 | 16 | 75 | 0.446 |
| Resorption[a] | 0/16 (0%) | 6/43 (14.0%) | 0/16 (0%) | | |
| Embryonic lethal[a] | 0/16 (0%) | 1/43 (2.3%) | 1/16 (6.3%) | | |
| Survivors[a] | 16/16 (100%) | 36/43 (83.7%) | 15/16 (93.8%) | 67 | 0.818 |
| Edema[b] | 0/16 (0%) | 1/36 (2.8%) | 5/15 (33.3%) | | |
| *Nsd2*$^{WT/-}$ intercross at E18.5 | *Nsd2*$^{+/+}$ | *Nsd2*$^{+/-}$ | *Nsd2*$^{-/-}$ | Total | p-value |
| Total embryos | 7 | 19 | 11 | 37 | 0.640 |
| Resorption[a] | 0/7 (0%) | 1/19 (5.3%) | 0/11 (0%) | | |
| Embryonic lethal[a] | 0/7 (0%) | 0/19 (0%) | 0/11 (0%) | | |
| Survivors[a] | 7/7 (100%) | 18/19 (94.7%) | 11/11 (100%) | 36 | 0.641 |
| Edema[b] | 0/7 (0%) | 1/18 (5.6%) | 1/11 (9.1%) | | |
| *Nsd2*$^{WT/P906L}$ intercross at E15.5 | *Nsd2*$^{WT/WT}$ | *Nsd2*$^{WT/P906L}$ | *Nsd2*$^{P906L/P906L}$ | Total | p-value |
| Total embryos | 34 | 74 | 21 | 129 | 0.067 |
| Resorption[a] | 3/34 (8.8%) | 18/74 (24.3%) | 1/21 (4.8%) | | |
| Embryonic lethal[a] | 1/34 (2.9%) | 0/74 (0%) | 0/21 (0%) | | |
| Survivors[a] | 30/34 (88.2%) | 56/74 (75.7%) | 20/21 (95.2%) | 106 | 0.329 |
| Edema[b] | 0/30 (0%) | 5/56 (8.9%) | 4/20 (20.0%) | | |
| *Nsd2*$^{WT/P906L}$ intercross at E18.5 | *Nsd2*$^{WT/WT}$ | *Nsd2*$^{WT/P906L}$ | *Nsd2*$^{P906L/P906L}$ | Total | p-value |
| Total embryos | 36 | 61 | 19 | 116 | 0.071 |
| Resorption[a] | 3/36 (8.3%) | 8/61 (13.1%) | 1/19 (5.3%) | | |
| Embryonic lethal[a] | 0/36 (0%) | 3/61 (4.9%) | 1/19 (5.3%) | | |
| Survivors[a] | 33/36 (91.7%) | 50/61 (82.0%) | 17/19 (89.5%) | 100 | 0.077 |
| Edema[b] | 0/33 (0%) | 0/50 (0%) | 1/17 (5.9%) | | |

Goodness-of-fit tests for the total embryos and survivors were performed at each embryonic stage.

[a]The numbers in brackets represented the percent of the abnormality to the total embryos in each corresponding genotype.

[b]The numbers in brackets represented the percent of the abnormality to the survivors in each corresponding genotype.

## Fetal growth retardation and enlargement of placentas in *Nsd2* mutant mice

Next, we assessed the effects of loss-of-function mutation of NSD2 on fetal growth. For *Nsd2*-knockout line, 18 *Nsd2*$^{+/+}$, 34 *Nsd2*$^{+/-}$, and 9 *Nsd2*$^{-/-}$ embryos were analyzed at E15.5, and 9 *Nsd2*$^{+/+}$, 13 *Nsd2*$^{+/-}$, and 11 *Nsd2*$^{-/-}$ embryos were analyzed at E18.5 (S1 Table). Although the average body weight of *Nsd2*$^{-/-}$ embryos at E15.5 was lower than that of *Nsd2*$^{+/+}$ and *Nsd2*$^{+/-}$ embryos, the difference was not statistically significant (p = 0.120 > 0.05) (Fig 1A, left panel). Meanwhile, the body weights of *Nsd2*$^{-/-}$ embryos at E18.5 were significantly lower than those of *Nsd2*$^{+/+}$ (p = 0.007) and *Nsd2*$^{+/-}$ (p = 0.031) embryos (Fig 1A, left panel). For *Nsd2*-knockin line, 24 *Nsd2*$^{WT/WT}$, 35 *Nsd2*$^{WT/P906L}$, and 11 *Nsd2*$^{P906L/P906L}$ embryos at E15.5, and 29 *Nsd2*$^{WT/WT}$, 47 *Nsd2*$^{WT/P906L}$, and 16 *Nsd2*$^{P906L/P906L}$ embryos at E18.5 were analyzed. The body weights of *Nsd2*$^{P906L/P906L}$ embryos were significantly lower than those of *Nsd2*$^{WT/WT}$ embryos at E15.5 (p = 0.030). The body weights of *Nsd2*$^{P906L/P906L}$ embryos were tended to lower than those of *Nsd2*$^{WT/WT}$ embryos at E18.5 (p = 0.125) (Fig 1B, left panels). The growth retardation was observed in *Nsd2* mutant embryos at late-gestation.

Then, we assessed the effects of loss-of-function mutation of NSD2 on placental growth. In contrast to the body weights, significant increase in placental weight were observed in *Nsd2*$^{-/-}$ and *Nsd2*$^{+/-}$ compared to those of the littermate *Nsd2*$^{+/+}$ at E15.5 (p < 0.01) but not at E18.5 (Fig 1A, middle panels). The ratio of placenta weight to body weight (placenta/ body weight ratios) was significantly higher in *Nsd2*$^{-/-}$ compared to *Nsd2*$^{+/+}$ at both E15.5 (p = 0.001) and E18.5 (p = 0.012) (Fig 1A, right panels). The placenta/body weight ratios were also significantly higher in the *Nsd2*$^{+/-}$ at E15.5 (p = 0.027) compared to *Nsd2*$^{+/+}$ (Fig 1A, right panel). Simultaneously, the placental weights of *Nsd2*$^{P906L/P906L}$ or *Nsd2*$^{WT/P906L}$ were significantly higher than those of the littermate *Nsd2*$^{WT/WT}$ at E15.5 (p < 0.01) and E18.5 (p < 0.001) (Fig 1B, middle panels). The placenta/body weight ratios were significantly higher in *Nsd2*$^{P906L/P906L}$ compared to *Nsd2*$^{WT/WT}$ at E15.5 (p < 0.001) and E18.5 (p < 0.001), and *Nsd2*$^{WT/P906L}$ at E15.5 (p = 0.030) and E18.5 (p = 0.001) (Fig 1B, right panels). The placenta/ body weight ratios were also significantly higher in *Nsd2*$^{WT/P906L}$ at E15.5 and E18.5 (p < 0.001) compared to *Nsd2*$^{WT/WT}$ (Fig 1B, right panels). These results demonstrated that the loss of function mutation of NSD2 causes placental enlargement (placentomegaly).

## Histological abnormalities observed in *Nsd2* mutant placentas

During mid-gestation, the mouse placenta completes its characteristic three-layered structure: the decidua, junctional zone, and labyrinthine zone. We compared the placental morphologies of homozygous mutant embryos and the wild-type controls at E15.5. The histological sections of six *Nsd2*$^{-/-}$ and six littermate *Nsd2*$^{+/+}$ placentas were stained with HE for general morphology and PAS to detect glycogen trophoblast and spongiotrophoblast cells in the junctional zone. We observed structural abnormalities of the labyrinth and junctional zones in *Nsd2*$^{-/-}$ placentas (Fig 2). The boundary between the two fetal layers was disorganized (Fig 2A). The thickness of the two fetal layers at the center of the placenta was measured for six *Nsd2*$^{-/-}$ and six *Nsd2*$^{+/+}$ littermates (Fig 2B). The labyrinth zone was significantly thicker in *Nsd2*$^{-/-}$ than in *Nsd2*$^{+/+}$ placentas (p = 0.037). There was a trend towards thickening of the junctional zone in the *Nsd2*$^{-/-}$ placentas but these measurements were not statistically significant (p = 0.094). In the labyrinth of *Nsd2*$^{-/-}$ placentas, the cells were sparse, in contrast to the dense distribution of cells in *Nsd2*$^{+/+}$ controls (Fig 2C). Maternal blood sinuses were dilated in the *Nsd2*$^{-/-}$ placentas, and the adjacent trophoblasts appeared atrophic with little cytoplasm (Fig 2C, magnified panels). Morphological abnormalities were also observed in the junctional zone of *Nsd2*$^{-/-}$ placentas. As shown in Fig 2D, enlargement of the junctional zone with an increase in the number of glycogen trophoblast and spongiotrophoblast cells was evident in one *Nsd2*$^{-/-}$ mutant placenta. Both trophoblasts tended to be larger in size in the *Nsd2*$^{-/-}$ placenta compared to the wild-type littermates (Fig 2D, magnified panels). Similar results were obtained in the other two *Nsd2*$^{-/-}$ mutant placentas (S1 Fig).

Trophoblast cells that are closest to the maternal blood sinuses in the labyrinth have been reported to exhibit alkaline phosphatase activity [15]. To describe the effect of *Nsd2* loss-of-function on labyrinth structure more accurately, we performed histochemical staining the placental sections of one *Nsd2*$^{-/-}$ and two *Nsd2*$^{+/+}$ littermates with alkaline phosphatase (S2 Fig). In *Nsd2*$^{+/+}$ placentas, intrinsic alkaline phosphatase activity was detected at various levels in trophoblast cells

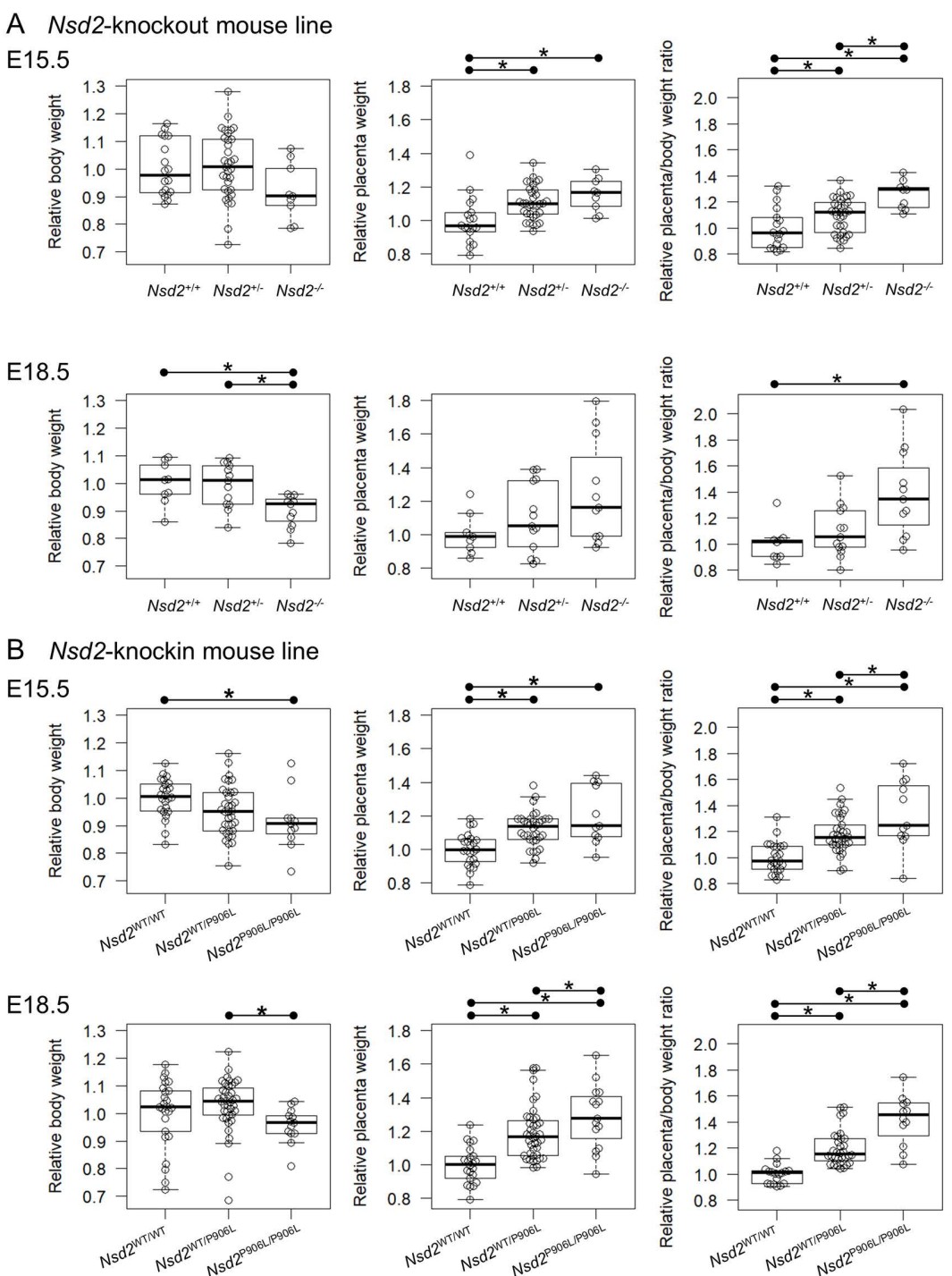

**Fig 1. Loss-of-function mutation of NSD2 causes fetal growth retardation and enlargement of placentas in mice.** Relative body weights (left panels), relative placenta weights (middle panels), and relative placenta/body weight ratios (right panels) are shown as box plots for three genotype groups of *Nsd2*-knockout (A) and *Nsd2*-knockin (B) embryos at E15.5 and E18.5. Each open dot represents the value for an individual sample. Asterisks indicate statistically significant differences (Holm-corrected, p<0.05) between the two groups, as indicated by the horizontal line segment above each panel. The p-values for the statistical tests are shown in S1 Table.

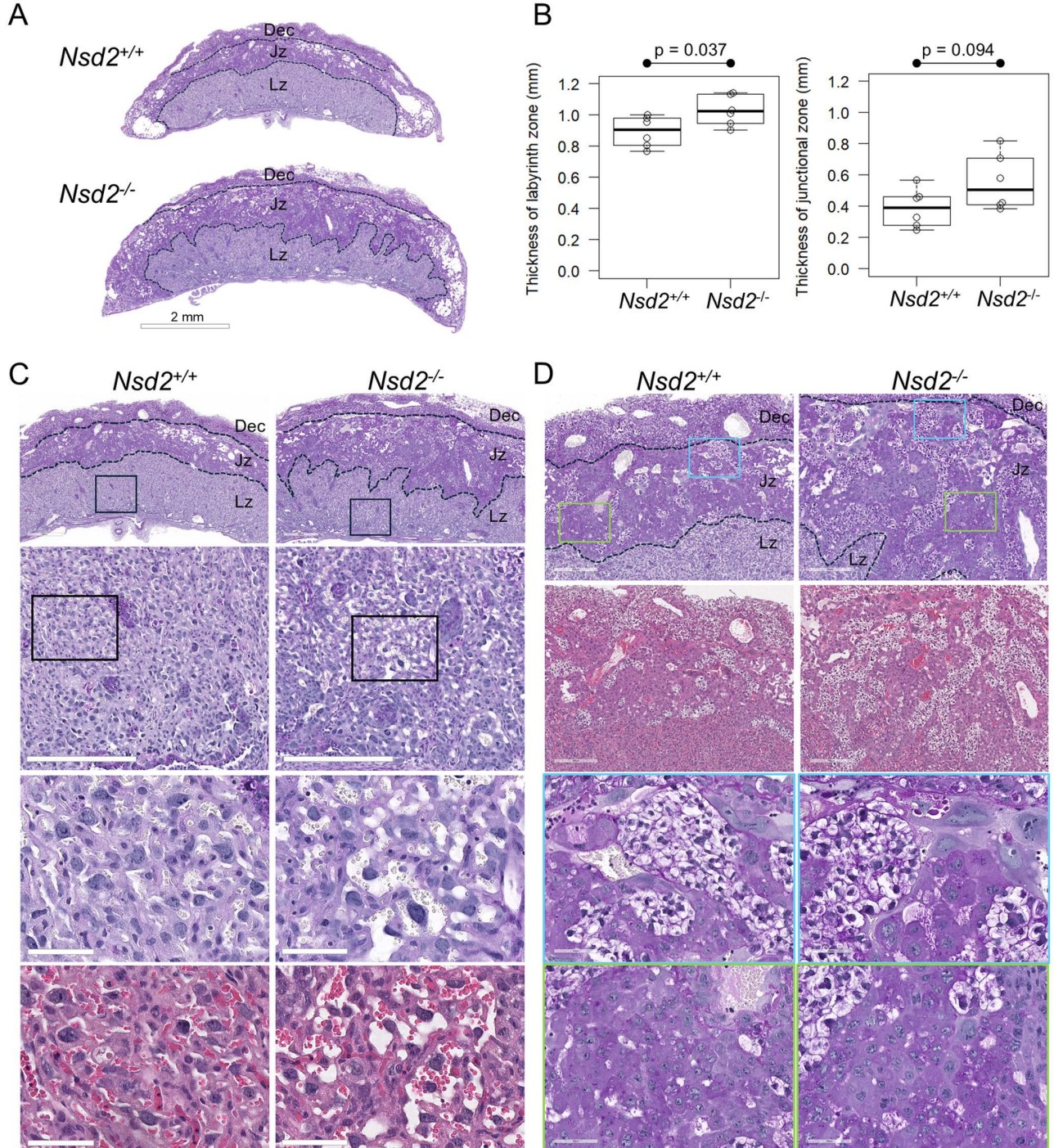

**Fig 2. Histological abnormalities observed in the *Nsd2*<sup>-/-</sup> mutant placentas at E15.5.** PAS-stained placental sections of *Nsd2*<sup>-/-</sup> and *Nsd2*<sup>+/+</sup> littermates are shown. (A) Structural abnormalities in the placentas. Scale bar = 2 mm. (B) Quantifying the thicknesses of labyrinth zone and junctional zone in *Nsd2*<sup>-/-</sup> and *Nsd2*<sup>+/+</sup> littermates (n = 6 for each genotype group). Statistical significance was assessed using a two-tailed paired Student's t-tests. (C) Morphological abnormalities in the labyrinth zone. The magnified images show dilated maternal sinuses in *Nsd2*<sup>-/-</sup> placentas. (D) Morphological abnormalities in the junctional zone. In *Nsd2*<sup>-/-</sup> placentas, enlarged glycogen trophoblast (which appears foamy) and spongiotrophoblast cells are observed. HE-stained serial sections are shown in C and D. Dashed lines indicate the boundaries of the three layers. Dec, decidua; Jz, junctional zone; Lz, labyrinth zone. The colored boxes represent the respective magnified areas shown below. Scales are from top to bottom: 600, 300, 60, and 60 μm in C; 300, 300, 60, and 60 μm in D.

at the maternal blood sinuses near the spongiotrophoblast border (S2A and S2C Figs) and the base (embryonic side) of the labyrinth zone (S2B and S2D Figs). In contrast to wild-type controls, markedly higher levels of alkaline phosphatase activity throughout the labyrinth zone (S2 top right, S2E and S2F Figs) and dilated maternal blood sinuses were evident in one Nsd2$^{-/-}$ mutant placenta (S2E Figs). These results confirm the morphological abnormalities of the labyrinth structure in Nsd2-knockout placentas.

For Nsd2-knockin line, two Nsd2$^{P906L/P906L}$ and three littermate Nsd2$^{WT/WT}$ placentas were stained with HE and PAS for histological analysis. As shown in Fig 3, similar morphological abnormalities of the fetal layers were observed as in the Nsd2$^{-/-}$ mutant placentas: the labyrinth and junctional zone boundaries of Nsd2$^{P906L/P906L}$ placentas were disorganized (Fig 3A), and the thickness of the two fetal layers in the center of placenta tended to be thicker than that of Nsd2$^{WT/WT}$ placentas (Fig 3B). Compared with the wild-type littermates, Nsd2$^{P906L/P906L}$ placentas showed dilated maternal blood sinuses in the labyrinth zone (Fig 3C) and expanded junctional zone with a slightly increase in the number and size of glycogen trophoblast and spongiotrophoblast cells (Fig 3D). All these results demonstrate that enlargement of placentas caused by the loss-of-function mutation of NSD2 is accompanied by morphological abnormalities at the layered structure and cellular levels. The observed morphological abnormalities suggest deteriorated functions of Nsd2$^{-/-}$ and Nsd2$^{P906L/P906L}$ placentas.

### Fetal abnormalities observed in Nsd2 mutant mice

We further observed prominent subcutaneous edema in five out of 15 Nsd2$^{-/-}$ embryos (33.3%) and four out of 20 Nsd2$^{P906L/P906L}$ embryos (20.0%) at E15.5 (Table 1 and Fig 4A). Subcutaneous edema was also observed in homozygous mutant embryos (Nsd2$^{-/-}$ and Nsd2$^{P906L/P906L}$) at E18.5 and in heterozygous embryos (Nsd2$^{WT/-}$ and Nsd2$^{WT/P906L}$) at lower frequencies but was not observed in wild-type embryos (Table 1). We also observed a ventricular septal defect (VSD) of the heart in one out of four Nsd2$^{P906L/P906L}$ embryos that exhibited subcutaneous edema at E15.5 (Fig 4B). The observed VSD in some but not all homozygous mutant embryos was consistent with the results of a previous study that reported a membranous VSD in half of Whsc1 (Nsd2)$^{-/-}$ embryos lacking exons 13–21 of Whsc1 at E18.5 [16]. Taken together with placental enlargement in Nsd2$^{-/-}$ and Nsd2$^{P906L/P906L}$ embryos, our results demonstrate that loss of function mutation of NSD2 leads to abnormal development in both the fetus and placenta during mid-gestation.

## Discussion

Our study demonstrated that the loss-of-function mutation of murine NSD2 causes placental enlargement (placentomegaly) accompanied by morphological changes during late-gestation. Nsd2 deletion or loss-of-function resulted in fetal growth retardation with varying severity among homozygous mutants. However, these are not necessarily embryonic lethal, which allowed us to elucidate Nsd2 effects on placental development at late-gestation. In mice, the formation of the three layers in the placenta (decidua, junctional zone, and labyrinth zone) is completed by E14.5. The glycogen storage, endocrine and transport functions of the murine placenta occur in structurally discrete junctional zone and labyrinth zone, respectively. The junctional zone contains spongiotrophoblast cells and glycogen cells and provides energetic (glycogen), hormonal and physical support to ensure correct placentation and pregnancy progression [17,18]. The labyrinth zone comprises fetal and maternal blood spaces separated by the fetal vascular endothelium and specialized trophoblast cell types. The labyrinth zone provides an interface for exchange between maternal and fetal circulation [19]. The major abnormalities in the enlarged placentas in Nsd2 mutant embryos are associated with expansion of the junctional zone due to increased size and number of glycogen cells and/or spongiotrophoblast cells, as well as dilation of the maternal blood sinuses in the labyrinth zone, which collectively disturb the architecture of the placental layers. A Mouse Placentation Spatiotemporal Transcriptomic Atlas spanning from embryonic day (E) 7.5 to E14.5 revealed that the Nsd2 gene is expressed more in inner ectoplacental cone, parietal trophoblast giant cells in the junctional zone, and labyrinth trophoblast progenitor cells at E7.5. However, once junctional glycogen trophoblast cells appeared at E10.5, Nsd2 was expressed at the highest levels in these cells compared to other cells [14]. The cell types in which Nsd2 is expressed are consistent with the cells in which we observed abnormal morphology in Nsd2$^{-/-}$ and Nsd2$^{P906L/P906L}$ placentas.

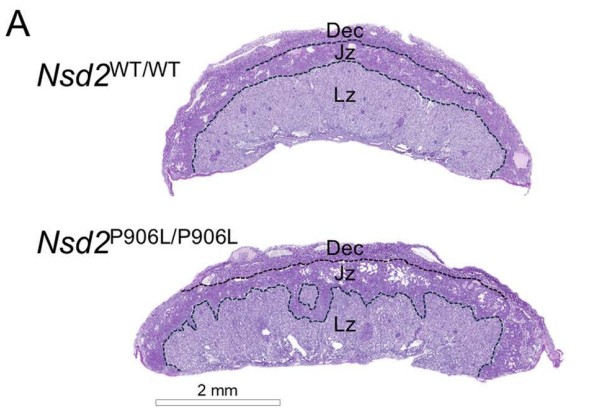

**B**

| Labyrinth zone [mm] | $Nsd2^{WT/WT}$ | $Nsd2^{P906L/P906L}$ |
|---|---|---|
| Littermate_1 | 1.000 | 1.139 |
| Littermate_2 | 0.851 | 0.943 |
| Junctional zone [mm] | $Nsd2^{WT/WT}$ | $Nsd2^{P906L/P906L}$ |
| Littermate_1 | 0.567 | 0.819 |
| Littermate_2 | 0.450 | 0.707 |

**Fig 3. Histological abnormalities observed in the $Nsd2^{P906L/P906}$ mutant placentas at E15.5.** PAS-stained placental sections of $Nsd2^{P906L/P906L}$ and $Nsd2^{WT/WT}$ littermates are shown. (A) Structural abnormalities in the placentas. Scale bar = 2 mm. (B) Quantifying the thicknesses of labyrinth and junctional zone in $Nsd2^{P906L/P906L}$ and $Nsd2^{WT/WT}$ littermates (n = 2 for each genotype group). (C) Morphological abnormalities in the labyrinth zone. The magnified images show dilated maternal sinuses in $Nsd2^{P906L/P906L}$ placentas. (D) Morphological abnormalities in the junctional zone. In $Nsd2^{P906L/P906L}$ placentas, enlarged glycogen trophoblast and spongiotrophoblast cells are observed. HE-stained serial sections are shown in C and D. Dashed lines indicate the boundaries of the three layers. The colored boxes represent the respective magnified areas shown below. Scales are from top to bottom: 600, 300, 60, and 60μm in C; 300, 300, 60, and 60μm in D.

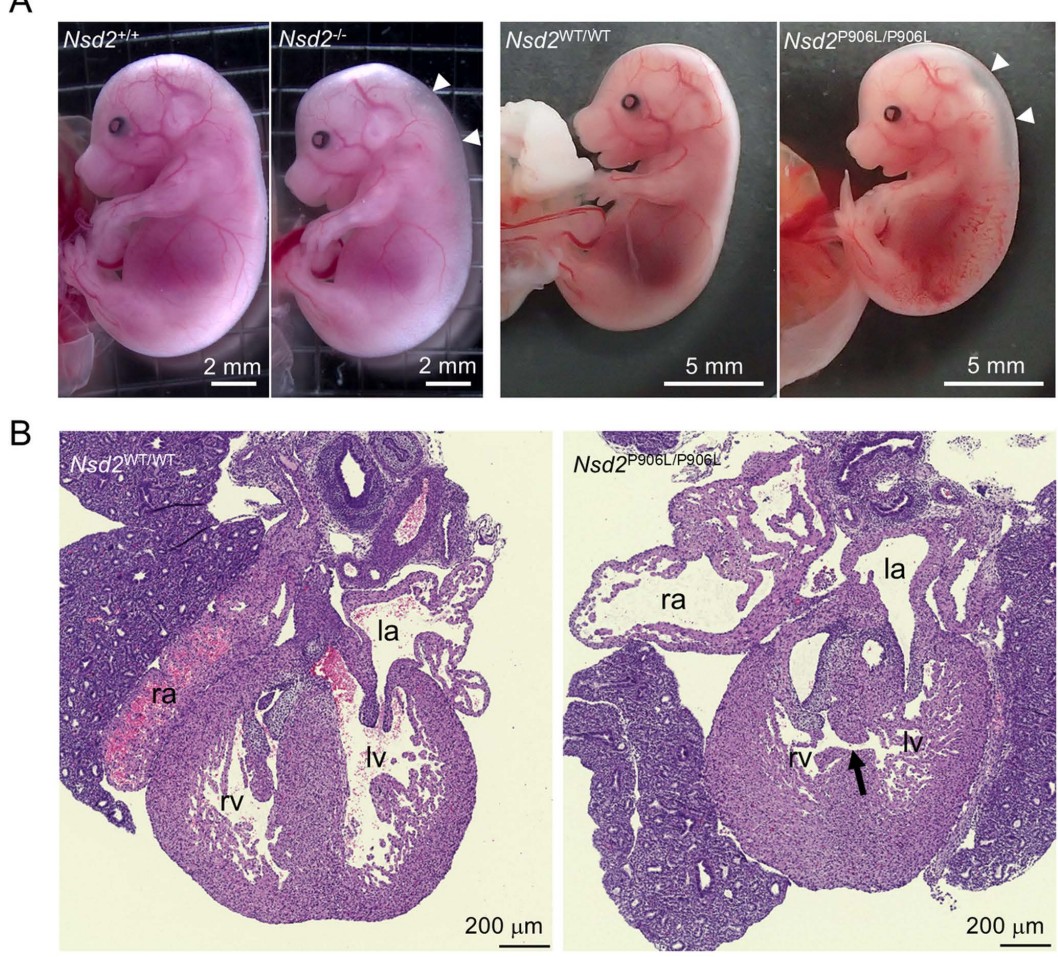

**Fig 4. Fetal abnormalities observed in *Nsd2* mutant embryos.** (A) Representative images of subcutaneous edema in the neck and back (white arrowheads) observed in *Nsd2*[-/-] and *Nsd2*[P906L/P906L] embryos at E15.5. The images of littermate wild-type embryos are also shown. (B) HE-stained heart sections of embryos at E15.5. The VSD observed in an *Nsd2*[P906L/P906L] embryo is indicated by a black arrow (right). A littermate *Nsd2*[WT/WT] heart section is shown (left). The left (la) and right (ra) atriums and left (lv) and right (rv) ventricles are indicated in the section images.

The enlargement and increased number of glycogen cells in the junctional zone suggest dysregulation of IGF-II signaling. The number of glycogen cells was reduced by more than half in the placentas of IGF-II null mice [20]. *Igf2* is an imprinted gene whose expression is controlled by DNA methylation. Imprinting defects are commonly associated with junctional zone abnormalities [17]. *ASCL2*, which is also regulated by genomic imprinting, is required for the early maintenance of glycogen trophoblast cells [21]. DNMT1o-deficient placentas show loss of DNA methylation at imprinted loci and altered expression of these genes, resulting in aberrant placental morphology with extension of the junctional zone into the labyrinth zone, similar to what we observed in *Nsd2*[-/-] and *Nsd2*[P906L/P906L] placentas [22]. Moreover, overexpression of *Ascl2* or knockout of *H19,* a maternally expressed gene, leads to increased placental glycogen storage and reduced fetal growth [23,24]. Similarly, the imprinted *Phlda2* knockout mice showed expansion of junctional zone with excessive glycogen accumulation, which reduced fetal growth [25]. *Plac1* (placental-specific 1) is an X-linked gene that is maternally expressed in trophoblast cells owing to imprinted X inactivation in the murine placenta and is expressed during placentation from E7.5 to E14.5 [26,27]. *Plac1*-deficient placentas have been demonstrated to exhibit erosion of

the junctional zone and dilatation of the maternal blood sinuses in the labyrinth zone. A recent study showed that loss of H3K27me3 at imprinting genome regions in the *Sfmbt2* microRNA cluster causes enlargement of cloned mouse placentas [28]. These studies suggest that dysfunction of the imprinted genes underlies the placentomegaly and fetal growth retardation in *Nsd2*-/- and *Nsd2*P906L/P906L mice. Further studies are needed to elucidate the DNA methylation status and/or expression levels of these imprinted genes in the placentas of *Nsd2* mutant mice. Additionally, it remains to be determined whether nutritional supply from increased glycogen cells in the junctional zone to the embryo is impeded. Meanwhile, edema in *Nsd2*-mutated fetuses might be related to dysregulation of glucose supply from the increased glycogen cells in *Nsd2*-mutated placentas.

Various epigenetic mechanisms underlie placental development and function, including histone modifications, DNA methylation, and crosstalk between these markers. DNA methylation is an epigenetic modification essential for growth, exemplified by embryonic and perinatal lethality in mice lacking de novo DNA methyltransferases (DNMTs) [29]. Andrews et al. [30] assessed the roles of DNMT3A, DNMT3B, and DNMT3L in gene regulation during murine placental development, demonstrating that the loss of *Dnmt3b* resulted in the derepression of germline genes in trophoblast lineages and impaired the formation of the maternal-fetal interface in the placental labyrinth. In this study, we observed marked dilation of the maternal blood sinuses in labyrinth of *Nsd2* mutant mice. In wild-type placentas, as the chorioallantoic interface undergoes more extensive branching during development, the trophoblast-lined sinusoid spaces become progressively smaller [15]. Taken together, these results would suggest disruption of labyrinth villus branching due to the loss-of-function of NSD2 in labyrinth trophoblasts. The disrupting labyrinth architecture in *Nsd2* mutant placentas would prevent the establishment of sufficient surface area for nutrient and gas exchange causing fetal growth restriction. Mutations in WNT and BMP family components display abnormalities of chorio-allantoic fusion of placenta, as well as defects in formation of the labyrinth vasculature, including failure of fetal vessels and the inappropriate formation of large edematous maternal blood spaces [17]. The dilation of the maternal blood sinuses in the labyrinth zone of *Nsd2* mutant mice may also involve abnormal WNT and BMP signaling. Especially, the early ExE-originating BMP4 signal is necessary for allantois and primordial germ cell (PGC) specification [31]. It is known that BMP4 signaling-related genes are downregulated by *Setdb1* knockdown during PGC-like cell [32]. SETDB1 catalyzes the tri-methylation of lysine 9 (K9) of histone H3. *Nsd2*-/- and *Nsd2*P906L/P906L may also dysregulate BMP signaling through H3K36me2 changes and/or DNA methylation changes or a completely different secondary regulation due to NSD2 dysfunction. Our results suggest a critical role for NSD2 in murine labyrinth morphogenesis in addition to the junctional zone morphological changes.

Radford et al. recently showed that syncytiotrophoblast defects commonly cause developmental heart diseases [33]. DNA methylation levels of certain genes in placental tissues are closely associated with fetal congenital heart disease in humans [34]. Although these studies indicate that placental abnormalities have been sporadically implicated as a source of developmental heart defects, the frequency with which the placenta is located at the root of congenital heart defects remains elusive. By adopting a trophoblast lineage-specific conditional knockout strategy for three genes (*Atp11a*, *Smg9*, and *Ssr2*) whose knockout resulted in placental and heart defects and embryonic lethality, the authors proved a strictly trophoblast-driven cause of conditional heart defects and embryonic lethality in one of the three lines (*Atp11a*) [33]. While our constitutional design of *Nsd2* knockout and knock-in mice did not allow us to determine whether the placental dysplastic phenotype observed in the embryos of *Nsd2*-/- and *Nsd2*P906L/P906L is causal for their growth retardation and neonatal death phenotypes, our results demonstrate the critical involvement of NSD2 in normal placental development. Conditional epiblast-specific knockout of progesterone immunomodulatory binding factor 1 (Pibf1), which leaves PIBF1 expression intact in trophectoderm-derived cells within the placental labyrinth, rescued the cardiac defects of Pibf1-null embryos. This suggests that the cardiovascular development of the embryo depends on PIBF1-mediated trophoblast syncytialization and placentation [35]. Meanwhile, the epiblast-specific knockout of *Inositol 1,4,5-trisphosphate receptor* family genes, which targets the genes in all the fetal tissues and extraembryonic mesoderm but not extraembryonic trophoblast cells, displayed embryonic lethality and placental defects, suggesting disruption of the fetal-maternal connection by fetal tissues

can induce placental dysfunction [36]. These results suggest that gene-dependent bidirectional interactions exist in the placenta-embryo axis. Hence, we cannot rule out the possibility that some placental changes, in *Nsd2* mutant mice are in part secondary to primary effects in the embryo. Applying trophoblast and embryonic lineage-specific conditional knockout strategies for *Nsd2* will further delineate the roles of NSD2 in fetal growth and mouse survival.

## Supporting information

**S1 Table. Statistical-test p-values in the comparisons of body and placental weights between wild-type and heterozygous/homozygous individuals.**
(TIF)

**S1 Fig. Morphological abnormalities observed in the junctional zone of *Nsd2*-/- mutant placentas at E15.5.** PAS-stained sections of *Nsd2*-/- and *Nsd2*WT/WT littermates are shown. The top figure shows a section of the same tissue as shown in Figure 2. Dashed lines indicate the boundaries of the three layers. gl, glycogen trophoblast cells (foamy appearance); sp, spongiotrophoblast cells. Sale bar = 200 µm.
(TIF)

**S2 Fig. Alkaline phosphatase histochemical staining in the labyrinth of *Nsd2*-/- placenta at E15.5.** One *Nsd2*-/- (id4) and two *Nsd2*+/+ (id3 and id5) littermate placentas were subjected to alkaline phosphatase staining. Boxes labeled with corresponding alphabets represent the enlarged areas shown in A-F. HE-stained serial sections are shown below each magnified image. Sale bar = 300 µm (top) and 100 µm (magnified images).
(TIF)

**S3 Fig. The expression levels of *Nsds* and *Dnmts* genes in mouse placenta.** Bar graphs show the cell type composition of gene expression using publicly available single-cell based spatial transcriptome data spanning from E7.5 to E14.5. [14].
(TIF)

## Acknowledgments

We acknowledge Professor Shuji Takada of the National Research Institute for Child Health and Development for generating the genome-edited mice. We thank the staff of the Division of Laboratory Animal Resources for animal care. We thank Editage (www.editage.com) for English language editing.

## Author contributions

**Conceptualization:** Eriko Ohnishi.

**Data curation:** Eriko Ohnishi.

**Formal analysis:** Eriko Ohnishi, Kazuhiko Nakabayashi.

**Funding acquisition:** Kenichiro Hata, Tomoko Kawai.

**Investigation:** Eriko Ohnishi, Shiori Kinoshita.

**Methodology:** Eriko Ohnishi, Kenichiro Hata, Tomoko Kawai.

**Resources:** Kenichiro Hata, Tomoko Kawai.

**Supervision:** Kenichiro Hata, Tomoko Kawai.

**Validation:** Eriko Ohnishi.

**Visualization:** Eriko Ohnishi, Kazuhiko Nakabayashi.

**Writing – original draft:** Eriko Ohnishi, Kazuhiko Nakabayashi.

**Writing – review & editing:** Eriko Ohnishi, Shiori Kinoshita, Kazuhiko Nakabayashi, Kenichiro Hata, Tomoko Kawai.

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
