## [Decision Letter · Decision Letter 0]

PONE-D-24-42707Loss-of-function of NSD2 causes abnormal placentation accompanied by fetal growth retardation in micePLOS ONE

Dear Dr. Kawai,

Thank you for submitting your manuscript to PLOS ONE. After careful consideration, we feel that it has merit but does not fully meet PLOS ONE’s publication criteria as it currently stands. Therefore, we invite you to submit a revised version of the manuscript that addresses the points raised during the review process.

Reviewers agreed that more in-depth analysis is warranted. Reviewer 1 was very specific while Reviewer 2’s comments were very general. As a practical matter all requests to clarify definitions, terminology and data presentation (mutations, strains, placental structure (“layers”) and cell types analyzed, observed phenotypes, etc.) must be addressed. This likely requires additional analyses but not necessarily additional experiments (although additional experiments would be considered). In addition, you must address the sentiment that additional clarification, discussion and interpretation are required to understand and evaluate the validity of this work in the context of preexisting knowledge. However, you needn’t address every reviewer comment in the same detail. The most important reviewer comments deal with placentomegaly and fetal growth restriction, a potential “placenta-heart-axis”, potential deviation from Mendelian ratios in survivors and the need for greater clarity and specificity of description of the placental changes observed.

In addition, the title should reflect the fact that the experimental perturbations are limited to genetic loss of and/or altered function (to be clarified). “Loss-of-function mutation of NSD2 is associated with …” rather than “Loss-of-function of NSD2 causes …”. Also, please discuss the possibility that some placental changes are entirely or in part secondary to primary effects in the embryo.

We look forward to receiving your revised manuscript.

Kind regards,

David S. Milstone

Academic Editor

PLOS ONE

Journal Requirements: When submitting your revision, we need you to address these additional requirements. 1. Please ensure that your manuscript meets PLOS ONE's style requirements, including those for file naming. The PLOS ONE style templates can be found at https://journals.plos.org/plosone/s/file?id=wjVg/PLOSOne_formatting_sample_main_body.pdf and https://journals.plos.org/plosone/s/file?id=ba62/PLOSOne_formatting_sample_title_authors_affiliations.pdf 2. We note that the grant information you provided in the ‘Funding Information’ and ‘Financial Disclosure’ sections do not match.  When you resubmit, please ensure that you provide the correct grant numbers for the awards you received for your study in the ‘Funding Information’ section.

Reviewers' comments:

Reviewer's Responses to Questions

**Comments to the Author**

1. Is the manuscript technically sound, and do the data support the conclusions?

Reviewer #1: No

Reviewer #2: Yes

2. Has the statistical analysis been performed appropriately and rigorously? 

Reviewer #1: Yes

Reviewer #2: Yes

3. Have the authors made all data underlying the findings in their manuscript fully available?

Reviewer #1: Yes

Reviewer #2: Yes

4. Is the manuscript presented in an intelligible fashion and written in standard English?

Reviewer #1: Yes

Reviewer #2: Yes

5. Review Comments to the Author

Reviewer #1: In this manuscript, the authors describe two separate mouse lines whereby the gene Nsd2 is knocked out (Nsd2-/-) or contains a point mutation (Nsd2^P906L). NSD2 is a histone methyltransferase and so presumably altering its expression/function will have implications for chromatin structure and gene regulation. The authors previously reported that both mutations lead to perinatal lethality at differing levels of penetrance and that at embryonic day (E)18.5, Nsd2-/- and Nsd2^P906L/P906L fetuses were growth restricted. This led the authors to consider how placenta structure is affected by these mutations in the late stages of gestation. They show an exciting placentomegaly phenotype associated with homozygosity for either Nsd2 mutation, which is an exciting and rare phenotype typically associated with models of epigenetic dysregulation. The authors also show a ventricular septal defect (VSD) in at least one mutant fetal heart. While there is substantial emerging data in the literature regarding the placenta-heart axis, this is not properly introduced by the authors and so the rationale for assessing the fetal heart in this context is unclear. The link between placentomegaly and a heart defect is novel and exciting, and this should be better discussed and even represented in the manuscript title. There are some issues in the description and characterisation of the placenta phenotype (listed below). Currently, the discussion is quite superficial without exploring the specific phenotypes observed in this unique context. However, the data in this study are promising and with a little more detail and finesse, this manuscript will contribute well to the literature.

Major concerns:

1. The authors do not indicate the effect of the Nsd2^P906L point mutation: does it knock out the enzyme function partially or completely? More information in the introduction is required so that a proper comparison between the Nsd2-/- and Nsd2^P906/P906L phenotypes can be achieved. Do the authors have any specific information about how histone methylation is affected by the Nsd2 mutations? Is the histone methylation of placenta or heart specific genes particularly affected? This would be a good addition to the manuscript.

Additionally, as the authors discuss the data throughout the manuscript, it would be helpful to the reader if the data from two mouse lines were kept separate. This might be in separate sentences or even separate paragraphs. This will allow the reader to digest the data and better determine the similarities and differences of the models. Also, the notations for the mutations should be consistent throughout the manuscript and clearer. The authors use the word ‘mutation’ and it is unclear which mouse line they are referring to (e.g., Lines 129-132), and, for example, use Nsd2^P906L/P906L and KI-hom terminology interchangeably. It would be particularly helpful to use the actual mutation name when labelling figures.

2. It is interesting that loss-of-function mutations in Nsd2 lead to perinatal lethality. Looking at the genotypic ratios at different stages of development, it is also possible that embryonic lethality of the homozygous conceptuses has occurred. For example, at E15.5, the following non-Mendelian ratios are reported in this manuscript for each genotype.

Nsd2+/+ : Nsd2+/- : Nsd2-/-, 2:4:1

Nsd2+/+ : Nsd2^+/P906L : Nsd2^P906L/P906L, 2:3:1

To fully appreciate this observation, the authors need to confirm that the genotyping procedure is working properly. This data is relevant to the manuscript because a severe placenta phenotype established through poor development and/or function can cause embryonic lethality. The authors should include this interpretation of the data when it is presented.

3a. The rationale for and description of the placenta phenotypes requires some work, and the current analysis is superficial. Is Nsd2 mRNA or protein expressed within the placenta? If so, in what cell types? This can be achieved by immunohistochemistry of Nsd2 on wildtype placenta sections at E15.5, or by assessing publicly available single cell transcriptome datasets obtained in wildtype placenta. The authors state in the discussion that Nsd2 is expressed in the placenta and cite Han et al, 2018 (ref 20) but do not explicitly indicate the cell types that Nsd2 is expressed within. Knowing its location of expression will help to justify why the analysis is focussed on a particular region of the placenta. The placentas from Nsd2-/- and Nsd2^P906L/P906L are clearly larger but the authors do not really explain why this is the case: more cells? bigger cells? which cells?

3b.The majority of the analysis in the manuscript focuses on the labyrinth layer, but it is also possible that the junctional zone of the placenta is also affected by Nsd2 loss-of-function, given its large size. The authors refer to this cell layer as the ‘spongiotrophoblast layer’, which is a slightly outdated term because this layer contains cells that are not spongiotrophoblast cells (e.g., parietal trophoblast giant cells, and glycogen trophoblast cells that play a role in glycogen storage and metabolism, and potentially other secretory roles (see Tunster et al, 2020 Reproduction PMID 32191912)). Defects in glycogen trophoblast cells associate with fetal growth defects and labyrinth defects. Currently, the function of glycogen trophoblast cells is not well understood, but presumably if the stored glycogen cannot be broken down into glucose for placenta and fetal use, fetal growth can be affected.

It is unclear why the authors have chosen to use PAS stain as the main histological stain to assess Nsd2-/- and Nsd2^P906L/P906L placentas. However, PAS stain can be used to identify glycogen trophoblast cells and their location (they are an invasive cell type appearing in the junctional zone and the decidua) and the approximate level of glycogen that is stored (darker pink PAS stain indicates more glycogen content). The authors should assess the glycogen trophoblast population at a higher magnification in their PAS stained histological sections to rule out a potential defect in this cell type. Are there more glycogen trophoblast or spongiotrophoblast cells, and is this why the junctional zone is larger in the mutants?

In the discussion (lines 203-6), the authors describe the placenta phenotype as ‘an erosion of the ST layer’, which is inconsistent with what is observed in Fig 2A and 2B whereby the junctional zone is much larger and not eroded. A better description might be that the boundary between the junctional zone and labyrinth layer is disorganised – however, based on experience, this type of dipping down of the junctional zone into the labyrinth can occur in the wildtype situation (see Simmons et al, 2008 BMC Genomics PMID: 18662396 for a clear example in Fig 4). It would be better for the authors to focus their discussion on the larger size of the junctional zone. What are reasons for the large size? Progenitor cell regulation? Increased cell proliferation? Size or number of cells dysregulated? Etc, etc.

3c. It is possible that there is a labyrinth layer defect, yet how this phenotype is described in the manuscript is concerning. First, the labyrinth structure shown in the control placenta images is not what is typically observed at E15.5 (Fig 2C). This can have a knock-on effect on how the mutant phenotype is described. Are the images from both Nsd2 wildtype placentas (P906L and knockout littermates) taken from the periphery of the placenta rather than in a central region? There is a noticeable lack of fetal vasculature in the labyrinth region particularly in the KO_WT placenta, which is unusual and unexpected for a control. (Are the images reversed for genotype??)

To describe the effect of Nsd2 loss-of-function on labyrinth structure more accurately, the authors should perform simple staining the histological sections with alkaline phosphatase (labels all cells lining the maternal blood sinusoids) and/or isolectin (labels fetal vascular endothelial cells). This will allow the authors to confidently identify and differentiate between maternal and fetal blood spaces. If the maternal blood sinusoids are indeed dilated as indicated in the manuscript and when compared to a proper control, it would suggest that labyrinth villus branching is impeded. In this case, the villi do not extend into the maternal blood sinusoids as they should and might result from a defect in branching morphogenesis. A good article that discusses this principle is Adamson et al, 2002 Dev Biol (PMID: 12376109).

Further characterisation of the effect of Nsd2 loss of function on the labyrinth requires assessment of specific cell subtypes. This can be achieved by immunostaining (e.g., MCT1 or MCT4) to label syncytriotrophoblast cells to determine if they are fusing together or elongating to form normal labyrinth architecture. Measuring nuclear size of sinusoidal trophoblast giant cells in the histological sections might indicate whether endoreduplication of the DNA occurred as normal. The authors should be more specific on lines 205-6 on how disrupting labryinth architecture can affect placenta function. If the labyrinth villi have a stunted structure, it would prevent the establishment of sufficient surface area for nutrient and gas exchange causing fetal growth restriction.

Reports in the literature that show impeded labyrinth villus structure are typically associated with smaller placentas, and so it is important for the authors to consider why their placentas are bigger in the discussion. The current discussion about the placenta is quite superficial and mechanism is not considered.

4. The introduction should contain information about the placenta-heart axis to provide rationale for the experiments and data in this manuscript. This is very interesting and novel dataset (placentomegaly + fetal growth restriction + VSDs), and should be a ‘selling point’ of the manuscript. I would also recommend altering the title to include the heart data. Perhaps something like: “Nsd2 loss-of-function mutations cause placentomegaly and disrupt fetal heart structure in mice”. More rationale or narrative is required at the start of Line 171 to discuss why the fetal heart is assessed here – potentially linking placenta phenotype with heart phenotype.

Were other less severe heart defects present in the mutant hearts that did not display VSD? Was the placenta phenotype more severe when associated with VSD?

5. The discussion section of this manuscript is superficial and does not sufficiently discuss the potential underlying mechanisms for the cause of placentomegaly in this and other models with an epigenetic regulatory link (Placentomegaly has been identified in several models: 1) Plac1 mutant, which is X-linked, 2) miR-127 mutant, which is located in the Rlt1 imprinted locus (see Ito et al, 2015 Development PMID: 26138447), 3) cloned embryos, which presumably is a process that alters the epigenome including at the Sfmbt2 miRNA cluster, 4) Nsd2 mutants, which is a histone methyltransferase). While I understand that the authors attempt to make this link, it is not outrightly stated (starting on line 208 would be a good place). What might be a good experiment to assess common mechanisms among these models including Nsd2 (e.g., assessing Plac1, miR-127, Sfmbt2 expression in the Nsd2 mutant placentas? Are these genes regulated by H3K36me2 in the placenta?).

The discussion of the Dnmt3b knockout manuscript (Andrews et al, 2023 PMID: 36690623) on lines 225-234 is relevant given that DNA hypomethylation occurs alongside labyrinth defects to cause fetal growth restriction. However, there is proportionately too much detail given about this mutant and this should be condensed and better related to the Nsd2 mutants. Why does a histone methyltransferase and DNA methyltransferase have a similar labyrinth phenotype but not the same placental phenotype? Similarly, the discussion of Radford et al, 2023 (PMID: 36859534) is extensive. Instead, the salient points about the placenta-heart axis should be emphasized along with other articles that discuss this principle.

Other issues:

- the y-axis in all graphs should start at zero as is standard practice.

- Lines 117-8: cite the previous results that you are referring to that show fetal growth restriction at E18.5 in these mutant mouse lines.

- lines 150-2: this sentence is confusing and contradictory. It might be more accurate to state that there was a trend towards thickening of the “junctional zone” in the KO-homs but these measurements were not statistically significant.

- Line 197, line 240: E15.5 to E18.5 is late gestation not mid-gestation

- Line 200: While spongiotrophoblast cells have an endocrine function, the junctional zone that contains spongiotrophoblast cells also has glycogen trophoblast cells that have a metabolic function (store and metabolise glycogen). More information about these cells can be found in the following review (Tunster et al, 2020 Reproduction PMID 32191912).

- Figure 2: it would be helpful to the reader to label directly on the placenta images in figure 2 including each placenta layer in panel A, and the region assessed in panel C.

Reviewer #2: This paper presents evidence that loss of function of NSD2, a marker of DNA methylation and transcriptional activity, results in fetal growth retardation and placental enlargement. The paper is a valuable contribution to the field and has potential for publication. Its strength lies in the careful observation of changes in placental morphology. However, the paper does not address the precise mechanism of methylation dysregulation underlying the observed morphological changes. A more comprehensive investigation and/or discussion of the mechanisms behind these changes would enhance the scientific rigor of the paper.

6. PLOS authors have the option to publish the peer review history of their article (what does this mean? ). If published, this will include your full peer review and any attached files.

**Do you want your identity to be public for this peer review?** For information about this choice, including consent withdrawal, please see our Privacy Policy .

Reviewer #1: No

Reviewer #2: No

---

## [Author Response · Author response to Decision Letter 1]

1 Mar 2025

PONE-D-24-42707

Title: Response − Loss-of-function mutation of NSD2 is associated with abnormal placentation accompanied by fetal growth retardation in mice (Original title on initial submission: Loss-of-function of NSD2 causes abnormal placentation accompanied by fetal growth retardation in mice)

Dear Editor,

Thank you for giving us the opportunity to revise our manuscript. Your valuable feedback helped us to improve our manuscript. We have thoroughly reviewed all the comments and have made appropriate revisions to address these points in the manuscript.

Editor’s comments:

Reviewers agreed that more in-depth analysis is warranted. Reviewer 1 was very specific while Reviewer 2’s comments were very general. As a practical matter all requests to clarify definitions, terminology and data presentation (mutations, strains, placental structure (“layers”) and cell types analyzed, observed phenotypes, etc.) must be addressed. This likely requires additional analyses but not necessarily additional experiments (although additional experiments would be considered).

Response: Thank you for bringing this to our attention. We have comprehensively addressed all requests to clarify definitions, terminology, and data presentation throughout the manuscript:

1. Cell types: We added new experiments and analyses to better define placental cell types, particularly glycogen trophoblast and spongiotrophoblast cells in the junctional zone.

2. Terminology: We standardized terminology throughout the manuscript, consistently using 'junctional zone' instead of 'spongiotrophoblast layer' and ensuring consistent naming of all placental structures.

3. Mutation descriptions: We improved the descriptions of both our Nsd2-/- knockout and Nsd2P906L/P906L point mutation models, clearly explaining their molecular consequences on protein function and histone modification in the revised Introduction.

4. Data presentation: We completely reorganized our data presentation to clearly separate results from our two gene-editing mouse lines, making comparative analyses more straightforward.

5. Phenotype descriptions: We refined descriptions of all observed phenotypes to be more precise, with particular attention to placental morphological changes.

All modifications are highlighted in the ‘Revised Manuscript with Track Changes’.

In addition, you must address the sentiment that additional clarification, discussion and interpretation are required to understand and evaluate the validity of this work in the context of preexisting knowledge. However, you needn’t address every reviewer comment in the same detail. The most important reviewer comments deal with placentomegaly and fetal growth restriction, a potential “placenta-heart-axis”, potential deviation from Mendelian ratios in survivors and the need for greater clarity and specificity of description of the placental changes observed.

Response: Thank you for clarifying the reviewer’s comments. We have discussed our results more thoroughly to evaluate the validity of this work in the context of preexisting knowledge.

Regarding placentomegaly and fetal growth restriction, previous studies have shown this coincidence in mice with dysregulated genomic imprinting. We believe similar epigenetic molecular mechanisms may underlie the phenotype in our Nsd2 dysfunction mice. Our study suggests that Nsd2 could act upstream in the epigenetic regulation of placental development.

Concerning a potential "placenta-heart-axis," we acknowledge limitations in describing the relationship between these two organs based on our conventional gene-edited null mice. However, we have discussed the possibility that placental dysfunction resulting from Nsd2 loss might influence embryonic heart development, consistent with previous studies showing that placental cell-specific gene manipulation induced heart abnormalities.

We have reanalyzed the genotype distribution as recruiting only embryos for which the genotypes of all of the littermates have been identified and found that the deviation from Mendelian ratios in survivors was not statistically significant.

Regarding clarity and specificity of description of the placental changes, as mentioned in the above response, we have comprehensively addressed this throughout the manuscript.

In addition, the title should reflect the fact that the experimental perturbations are limited to genetic loss of and/or altered function (to be clarified). “Loss-of-function mutation of NSD2 is associated with …” rather than “Loss-of-function of NSD2 causes …”.

Response: Thank you for this suggestion. We agree that our title should more accurately reflect the nature of our findings. We will change the title from 'Loss-of-function of NSD2 causes...' to 'Loss-of-function mutation of NSD2 is associated with...' to better represent the relationship we've observed without overstating causality.

Also, please discuss the possibility that some placental changes are entirely or in part secondary to primary effects in the embryo.

Response: Thank you for pointing out this. we acknowledge this important consideration. In our revised discussion, we have added a section addressing this possibility. While our data shows clear placental abnormalities in NSD2 mutant mice, we cannot definitively rule out that some of these changes might be secondary responses to primary embryonic defects. This is particularly relevant considering the complex bidirectional signaling between placenta and embryo during development. We have included this alternative interpretation and discussed how future studies using tissue-specific knockouts could help distinguish between primary and secondary effects.

Response to Reviewer 1’s comments:

Major concerns:

1 The authors do not indicate the effect of the Nsd2^P906L point mutation: does it knock out the enzyme function partially or completely? More information in the introduction is required so that a proper comparison between the Nsd2-/- and Nsd2^P906/P906L phenotypes can be achieved. Do the authors have any specific information about how histone methylation is affected by the Nsd2 mutations? Is the histone methylation of placenta or heart specific genes particularly affected? This would be a good addition to the manuscript.

Response: Thank you for this important question. Our previous studies [ref 8, 9] confirmed that both Nsd2-/- and Nsd2^P906L/P906L mice exhibit reduced Histone H3 K36 dimethylation (H3K36me2), which is accompanied by DNA hypomethylation. Moreover, we previously demonstrated that the NSD2 P906L substitution destabilizes the NSD2 protein both in vivo and in vitro. This suggests that the observed reduction in H3K36me2 in the Nsd2^P906L/P906L mice is primarily caused by decreased NSD2 protein levels rather than by altered enzymatic activity of the mutant protein.

We have added the effect of the Nsd2^P906L point mutation and considerable information about the differences between Nsd2-/- and Nsd2^P906L/P906L mice in the revised "Introduction" (lines 57 to 67):

“ Using genome editing techniques, we previously generated two mouse lines: an Nsd2-knockout line created by completely deleting Nsd2 genes (Nsd2-/-) [8,9], and a knock-in line carrying a patient-derived single nucleotide point mutation that results in a proline-to-leucine substitution at position 906 of NSD2 (NP_001074571.2) (Nsd2P906L/P906L) [8]. We demonstrated that this variant was pathogenic and showed that it destabilizes the NSD2 protein [8]. This finding aligns with recent research showing that 60% of pathogenic missense variants, among more than 500,000 variants across over 500 human protein domains, reduce protein stability [10]. Both Nsd2-/- and Nsd2P906L/P906L reduced H3K36me2 level, which accompanies DNA hypomethylation [8]. The genetic differences between Nsd2-/- and Nsd2P906L/P906L include the production of diverse mRNA isoforms of Nsd2 and the expression of non-coding RNAs present in the deleted region in Nsd2-/-.”

Regarding the specific histone methylation patterns in placenta or heart-specific genes, we acknowledge this is an important direction for further research. Investigation of histone methylation and epigenetically affected genes in Nsd2-mutated placenta is a critical subject for future studies and will likely provide valuable insights into the molecular mechanisms underlying the observed phenotypes.

1. Additionally, as the authors discuss the data throughout the manuscript, it would be helpful to the reader if the data from two mouse lines were kept separate. This might be in separate sentences or even separate paragraphs. This will allow the reader to digest the data and better determine the similarities and differences of the models. Also, the notations for the mutations should be consistent throughout the manuscript and clearer. The authors use the word ‘mutation’ and it is unclear which mouse line they are referring to (e.g., Lines 129-132), and, for example, use Nsd2^P906L/P906L and KI-hom terminology interchangeably. It would be particularly helpful to use the actual mutation name when labelling figures.

Response: Thank you for pointing this out. In the revised manuscript, we have kept the results from the two mouse lines separate. And we have made the notations for the mutations consistent and clearer throughout the manuscript and clearer.

2. It is interesting that loss-of-function mutations in Nsd2 lead to perinatal lethality. Looking at the genotypic ratios at different stages of development, it is also possible that embryonic lethality of the homozygous conceptuses has occurred. For example, at E15.5, the following non-Mendelian ratios are reported in this manuscript for each genotype.

Nsd2+/+ : Nsd2+/- : Nsd2-/-, 2:4:1

Nsd2+/+ : Nsd2^+/P906L : Nsd2^P906L/P906L, 2:3:1

To fully appreciate this observation, the authors need to confirm that the genotyping procedure is working properly. This data is relevant to the manuscript because a severe placenta phenotype established through poor development and/or function can cause embryonic lethality. The authors should include this interpretation of the data when it is presented.

Response: Thank you for pointing out critical things. In the first manuscript submitted, the number of embryos for which weighing was available was reported. In this revised table 1, to investigate the genotype ratio of embryos at the late-gestation, we have shown the number of embryos for which the genotypes of all of the littermates have been identified. Statistical assessment showed no significant deviations from Mendelian ratios between genotypes. The results are shown in the revised Table 1. The rate of embryonic lethality did not differ significantly between genotypes. Meanwhile, the rate of resorption was enriched in heterozygous fetus both in Nsd2+/- and Nsd2WT/P906L, although we cannot rule out the possibility of maternal tissue contamination under the isolation of cohesive placenta. The percentage of resorption was higher in wildtype than homozygous Nsd2-/- or Nsd2P906L/P906L. Taken together, a loss-of-function mutation in NSD2 is not necessarily embryonic lethal and does not significantly increase embryonic lethality. We described this in lines 130 to 144:

“To confirm the effects of loss-of-function mutation in NSD2 on development, we examined the genotypic ratios of progenies from intercrosses between heterozygous mutant mice in each Nsd2-knockout and -knockin line (Table 1). In Nsd2+/- intercrosses, both Nsd2-/- and Nsd2+/- embryos were present at the expected frequencies at both E15.5 and E18.5 stages. Embryonic lethality was observed at E15.5 in one out of 43 Nsd2+/- (2.3%) and one out of 16 Nsd2-/- (6.3%) embryos but not in the Nsd2+/+. No embryonic lethality was observed at E18.5 in Nsd2+/- intercrosses. In Nsd2WT/P906L intercrosses, embryonic lethality was observed in one out of 34 Nsd2WT/WT embryos (2.9%) at E15.5 and in three out of 61 Nsd2WT/P906L (4.9%) and one out of 19 Nsd2P906L/P906L (5.3%) embryos at E18.5. The surviving number of Nsd2WT/P906L and Nsd2P906L/P906L embryos tended to be less than the expected Mendelian ratios, especially at E18.5, but were not statistically significant (Table 1). Hence, a loss-of-function mutation in NSD2 is not necessarily embryonic lethal and does not significantly increase embryonic lethality. Resorption was detected more frequently in heterozygous progeny of both Nsd2-mutant lines, but the possibility of contamination by maternal tissue during dissection cannot be ruled out. A more detailed analysis is needed for an accurate assessment.”

3. 3a. The rationale for and description of the placenta phenotypes requires some work, and the current analysis is superficial.

Response: The following is our answer to number 3. It is divided into several points.

Is Nsd2 mRNA or protein expressed within the placenta? If so, in what cell types? This can be achieved by immunohistochemistry of Nsd2 on wildtype placenta sections at E15.5, or by assessing publicly available single cell transcriptome datasets obtained in wildtype placenta. The authors state in the discussion that Nsd2 is expressed in the placenta and cite Han et al, 2018 (ref 20) but do not explicitly indicate the cell types that Nsd2 is expressed within. Knowing its location of expression will help to justify why the analysis is focussed on a particular region of the placenta. The placentas from Nsd2-/- and Nsd2^P906L/P906L are clearly larger but the authors do not explain why this is the case: more cells? bigger cells? which cells?

Response: Thank you for your pointing. We added the information regarding how much and which cell types in normal mouse placenta expressed Nsd2 in the revised supplementary figure 3, referring to a recent study of the spatiotemporal transcriptomic atlas of mouse placentation of Wu et al, 2024 (revised ref # 14). Including this information, we discussed our results as considering Nsd2 expressing cell types (lines 298 to 305):

“A Mouse Placentation Spatiotemporal Transcriptomic Atlas spanning from embryonic day (E) 7.5 to E14.5 revealed that the Nsd2 gene is expressed more in inner ectoplacental cone, parietal trophoblast giant cells in the junctional zone, and labyrinth trophoblast progenitor cells at E7.5. However, once junctional glycogen trophoblast cells appeared at E10.5, Nsd2 was expressed at the highest levels in them compared to other cells [14]. The cell types in which Nsd2 is expressed are consistent with the cells in which we observed abnormal morphology in Nsd2-/- and Nsd2P906L/P906L placentas.”

3 3b. The majority of the analysis in the manuscript focuses on the labyrinth layer, but it is also possible that the junctional zone of the placenta is also affected by Nsd2 loss-of-function, given its large size. The authors refer to this cell layer as the ‘spongiotrophoblast layer’, which is a slightly outdated term because this layer contains cells that are not spongiotrophoblast cells (e.g., parietal trophoblast giant cells, and glycogen trophoblast cells that play a role in glycogen storage and metabolism, and potentially other secretory roles (see Tunster et al, 2020 Reproduction PMID 32191912)). Defects in glycogen trophoblast cells associate with fetal growth defects and labyrinth defects. Currently, the function of glycogen trophoblast cells is not well understood, but presumably if the stored glycogen cannot be broken down into glucose for placenta and fetal use, fetal growth can be affected.

Response: Thank you for highlighting these important points. We have corrected the terminology, as changing 'spongiotrophoblast layer' to 'junctional zone' throughout the manuscript.

3 3b. It is unclear why the authors have chosen to use PAS stain as the main histological stain to assess Nsd2-/- and Nsd2^P906L/P906L placentas. However, PAS stain can be used to identify glycogen trophoblast cells and their location (they are an invasive cell type appearing in the junctional zone and the decidua) and the approximate level of glycogen that is stored (darker pink PAS stain indicates more glycogen content). The authors should assess the glycogen trophoblast population at a higher magnification in their PAS stained histological sections to rule out a potential defect in this cell type.

---

## [Decision Letter · Decision Letter 1]

PONE-D-24-42707R1Loss-of-function mutation of NSD2 is associated with abnormal placentation accompanied by fetal growth retardation in micePLOS ONE

Dear Dr. Kawai,

Thank you for submitting your manuscript to PLOS ONE. After careful consideration, we feel that it has merit but does not fully meet PLOS ONE’s publication criteria as it currently stands. Therefore, we invite you to submit a revised version of the manuscript that addresses the points raised during the review process.

**Statistical analyses require further clarification as detailed below.**

We look forward to receiving your revised manuscript.

Kind regards,

David S. Milstone

Academic Editor

PLOS ONE

**Journal Requirements:**

**Additional Editor Comments:**

The authors have carefully addressed most of the criticisms of the reviewers and the editor. However, statistical analysis requires further clarification. The underlying issue appears to be that pairwise ratios (that reduce the variation in the control “values” to zero) appear to have been used but are not warranted by the experimental design. However, the method used is not clearly described so some ambiguity remains. The individual experimental and individual control observations/values do not appear to be methodologically paired in any empirically meaningful way. Without this built into the experimental design, paired values cannot be used to determine ratios. Instead, ratios must be determined based on the control values as a group and the variability of the control values (the ratios of individual control values to the mean of all control values) must be included in the statistical analyses. One way to do this is to calculate the ratios of the experimental and the control values to the mean of the control values. The experimental and control ratios can then be used to test null hypotheses in a statistically valid manner.

Description of the statistical analysis does not unambiguously state whether paired or non-paired values were used. Doing so will help clarify which statistical analyses are appropriate

Reviewers' comments:

Reviewer's Responses to Questions

**Comments to the Author**

1. If the authors have adequately addressed your comments raised in a previous round of review and you feel that this manuscript is now acceptable for publication, you may indicate that here to bypass the “Comments to the Author” section, enter your conflict of interest statement in the “Confidential to Editor” section, and submit your "Accept" recommendation.

Reviewer #2: All comments have been addressed

2. Is the manuscript technically sound, and do the data support the conclusions?

Reviewer #2: Yes

3. Has the statistical analysis been performed appropriately and rigorously? 

Reviewer #2: Yes

4. Have the authors made all data underlying the findings in their manuscript fully available?

Reviewer #2: Yes

5. Is the manuscript presented in an intelligible fashion and written in standard English?

Reviewer #2: Yes

6. Review Comments to the Author

**Reviewer #2: ** The author has meticulously addressed the reviewer's comments and has submitted a revised paper that is deemed satisfactory. I have no further comments.

7. PLOS authors have the option to publish the peer review history of their article (what does this mean? ). If published, this will include your full peer review and any attached files.

**Do you want your identity to be public for this peer review?** For information about this choice, including consent withdrawal, please see our Privacy Policy .

Reviewer #2: No

---

## [Author Response · Author response to Decision Letter 2]

6 Jun 2025

Dear Editor,

Thank you for giving us the opportunity to revise our manuscript. Your valuable feedback helped us to improve our manuscript. We have thoroughly reviewed all the comments and have made appropriate revisions to address these points in the manuscript.

Editor’s comments:

The authors have carefully addressed most of the criticisms of the reviewers and the editor. However, statistical analysis requires further clarification. The underlying issue appears to be that pairwise ratios (that reduce the variation in the control “values” to zero) appear to have been used but are not warranted by the experimental design. However, the method used is not clearly described so some ambiguity remains. The individual experimental and individual control observations/values do not appear to be methodologically paired in any empirically meaningful way. Without this built into the experimental design, paired values cannot be used to determine ratios. Instead, ratios must be determined based on the control values as a group and the variability of the control values (the ratios of individual control values to the mean of all control values) must be included in the statistical analyses. One way to do this is to calculate the ratios of the experimental and the control values to the mean of the control values. The experimental and control ratios can then be used to test null hypotheses in a statistically valid manner.

Description of the statistical analysis does not unambiguously state whether paired or non-paired values were used. Doing so will help clarify which statistical analyses are appropriate.

Response: Thank you for pointing these critical issues to our attention. We have clearly described the method for determination of ratios in the statistical analyses. We calculated the ratios of the experimental and the control values to the mean of the control values in this revised manuscript. Subsequently, we implemented a non-paired Wilcoxon rank sum test between the groups. Pairwise comparisons were performed between the three genotypes. We described these methods in lines 104 to 107 in “Mouse phenotyping and statistical analysis” in a revised manuscript.

---

## [Editor Report · Decision Letter 2]

Loss-of-function mutation of NSD2 is associated with abnormal placentation accompanied by fetal growth retardation in mice

PONE-D-24-42707R2

Dear Dr. Kawai,

We’re pleased to inform you that your manuscript has been judged scientifically suitable for publication and will be formally accepted for publication once it meets all outstanding technical requirements.

Kind regards,

David S. Milstone

Academic Editor

PLOS ONE

---

## [Editor Report · Acceptance letter]

PONE-D-24-42707R2

PLOS ONE

Dear Dr. Kawai,

I'm pleased to inform you that your manuscript has been deemed suitable for publication in PLOS ONE. Congratulations! Your manuscript is now being handed over to our production team.

Kind regards,

on behalf of

Dr. David S. Milstone

Academic Editor

PLOS ONE